# The Role of Folate Deficiency as a Potential Risk Factor for Nontraumatic Anterior Spinal Artery Syndrome in an Adolescent Girl

**DOI:** 10.3390/brainsci12111470

**Published:** 2022-10-29

**Authors:** Chun-Chieh Hu, Yung-Yu Yang, G. W. Gant Luxton, Yu-Pang Lin, Kuo-Sheng Hung, Chih-Fen Hu

**Affiliations:** 1Department of Pediatrics, Tri-Service General Hospital, National Defense Medical Center, Taipei 11490, Taiwan; 2Department of General Medicine, Tri-Service General Hospital, National Defense Medical Center, Taipei 11490, Taiwan; 3Department of Molecular and Cellular Biology, University of California-Davis, Davis, CA 95616, USA; 4Department of Radiology, Tri-Service General Hospital, National Defense Medical Center, Taipei 11490, Taiwan; 5Center for Precision Medicine and Genomics, Tri-Service General Hospital, National Defense Medical Center, Taipei 11490, Taiwan

**Keywords:** folate deficiency, nontraumatic anterior spinal artery syndrome, young stroke, acute flaccid paralysis

## Abstract

Nontraumatic anterior spinal artery syndrome (ASAS) is an extremely rare clinical condition in pediatric populations with a mostly unknown underlying etiology. Here we discuss the case of a previously healthy 14-year-old girl presenting with sudden onset acute flaccid paralysis to the emergency department. A spinal STIR/DWI MRI revealed hyperintensities extending from cervical vertebrae C3-6, consistent with the diagnosis of ASAS. In order to determine any precipitating causes of ASAS, we also extensively investigated established potential risk factors for ASAS in our patient and noticed that she had a marked folate deficiency requiring folic acid supplementation to prevent future episodes of ASAS as well as to repair the patient’s injured spinal cord. Interestingly, the patient did not display elevated levels of homocysteine nor did she possess the three pathogenic *MTHFR* mutations characteristic of ASAS. Although her folate deficiency did not cause responsive hyperhomocysteinemia, and she did not have pathogenic *MTHFR* mutations that impair the function of methylenetetrahydrofolate reductase in folate cycle, we suggest that isolated folate deficiency may play a role in adolescent cases of ASAS that, once identified, would require prompt identification and early intervention to improve the prognosis of these patients.

## 1. Introduction

Spinal cord infarction (SCI) is rare, accounting for approximately 0.3% to 1% of ischemic strokes and constituting only 1% to 2% of all neurological vascular emergencies [1]. Anterior spinal artery syndrome (ASAS) is an exceptionally rare acute ischemic spinal cord infarction, commonly presenting with an acute and painful myelopathy, resulting from the occlusion or hypoperfusion of the anterior spinal artery’s (ASA) blood supply, which consequentially impedes the circulation of the anterior two-thirds of the spinal cord [2]. The typical neurological symptoms of ASAS include acute loss of pain/temperature sensation and motor function below the affected levels of the ischemic spinal cord, which is often accompanied by neurogenic bowel/bladder dysfunction. ASAS is extremely rare in children, given the abundant collateral blood supply within their young spinal cords. Notably, spinal cord injuries without radiographic abnormality (SCIWORA), also known as ‘pediatric syndrome of traumatic myelopathy without demonstrable vertebral injury’, is one of the important differential diagnoses in ASAS. SCIWORA is mostly caused by hyperextension or flexion injuries due to characteristic anatomic differences of the more deformable and more elastic spine in children. These flexion–extension injuries, especially at the cervical spinal cord, may induce a transient occlusion of the vertebral arteries or the ASA with the result of an SCI [3]. While some cases of ASAS could be attributed to a vertebral fracture [4], spondylosis [2,5,6], and/or some minor trauma [4] before ASAS onset, a large proportion of cases may not have a definite cause. Although a few studies have tried to investigate the etiology or risk factors of ASAS, the identification of nontraumatic ASAS etiology in children remains a challenging topic needing further investigation since a large population of these patients remain idiopathic even after an extensive workup [7]. Here, we present one adolescent case of ASAS where we discovered, during our serial workup, that she had profound folate deficiency. We have also provided a focused review of the role of folate deficiency in ischemic stroke and recruited similar cases from external references for a summarized comparison and discussion. 

## 2. Case Presentation

A morbidly obese but relatively healthy and energetic 14-year-old right-handed female presented to the emergency department with a sudden onset of bilateral numbness and weakness in her arms, which spread downward into her legs. Before the onset of this episode, the patient was in a stable sitting position. She was alert, oriented, and verbally fluent. We inquired about any trauma history, even minor or trivial injury or posture (hyperextension or flexion), but she could not recall any relevant events that can cause this condition. Her physical examination showed intact cranial nerve function with relatively stable vitals. However, an unequal muscle weakness was observed in the patient’s four limbs, especially in the upper right and lower left extremities, which displayed muscle strength scores (Medical Research Council Scale, MRC scale) of 2/5 and 1/5, respectively. The patient’s sensory exam demonstrated that she had diminished responses to pain, temperature, and crude touch. Yet, proprioception and responses to vibration and fine touch were unaffected. An increased patellar reflex in the lower extremities, alongside a positive Babinski reflex on the right side, were also noted. However, no organic lesions in the patient’s cerebrum were detected by emergent contrast-enhanced brain computed tomography (CT) with CT angiography/venography reconstruction. While in a state of acute flaccid paralysis, the patient was admitted to the intensive care unit for further evaluation and management.

Soon after admission, a lumbar puncture was performed on the patient to harvest cerebral spinal fluid (CSF) to rule out the presence of infection and/or inflammation in her central nervous system. The patient’s biochemistry, cell counts, Gram staining, FilmArray® (Salt Lake City, UT, USA) meningitis/encephalitis (M/E) panel, and CSF culture were all negative. Despite an overall unremarkable cerebral CT and magnetic resonance image (MRI), hyperintensities extending from the cervical vertebrae C3-6 (Figure 1) were still observed in the short tau inversion recovery/diffusion weighted imaging (STIR/DWI) MRI, which is consistent with the possible ASAS present in the patient [8,9]. A young stroke panel was also used to extensively survey for hematological conditions (protein C, protein S, anti-thrombin, PT/aPTT/INR, and D-dimer), autoimmune diseases [10] (anti-ds DNA, anti-Cardiolipin IgG/IgM, ANA titer/pattern, anti-β2 glycoprotein, C3, C4, LA(DRVVT), anti-AQP4 IgG, and anti-MOG IgG), metabolic statuses (T3, T4, free T4, TSH, anti-TPO, HbA1C, plasma Lyso-Gb3, HDL, LDL, triglyceride, uric acid, total protein, vitamin B12, folate, and homocysteine), and infection issues (HIV Ag/Ab and RPR). Except for low serum folate levels (1.7 ng/mL, reference range: 4.8–37.3) and mildly increased LDL cholesterol levels (114 mg/dL, reference range: <100), no other abnormalities were identified in the patient. The patient received a five-day course of high-dose intravenous methylprednisolone to ameliorate the spinal cord edema, aspirin for the anticoagulation [11], and folic acid supplementation for the marked folate deficiency (Table 1). When folate levels were rechecked two weeks later, they were restored to normal levels (12.2 ng/mL). We also tested the patient’s genome for the presence of three pathogenic variants of the methylenetetrahydrofolate reductase (*MTHFR*) gene (NM_005957.5: c.1969A>G, p.Ter657Arg (rs768434408); c.1304_1305del, p.Phe435fs (rs4846051); c.470C>T, p.Arg157Gln (rs121434295)), but none were detected. 

On the other hand, her spinal MRI revealed mild herniated intervertebral disc (HIVD), but no bone marrow edema, no obvious structural abnormality at the spine, and the vertebra was found and the chance of fibrocartilaginous embolism was low, so we did not perform spinal digital subtraction angiography (DSA) and magnetic resonance angiogram (MRA) in this case. Although the possibility of minor trauma was low in the initial evaluation, no advanced images to investigate the offending pathology to the ASA could be a potential factor that limits the value of this study. Additionally, a 96-day follow-up spinal MRI revealed a myelomalacia extending from cervical vertebrae C4–C5 from a previous infarction (Figure 1). A 20-week laboratory workup also indicated that the patient’s folate levels dropped back down to 4.2 ng/mL. Thus, the patient was again supplemented with folic acid beginning at her 24-week follow-up appointment. The patient’s initial modified Rankin Scale (mRS), a measure of neurologic disability, of 5/6 gradually improved to 1/6 by her final outpatient visit. Additionally, the patient’s self-evaluation score improved from 20/100 to 92/100 within 6 months of treatment. She can go to school and handle her daily life without assistance. Furthermore, she is able to play volleyball as usual even though she felt a bit clumsy and presented mildly impaired coordination. She once mentioned that she started to feel unbearable pain over her limbs, so we added pregabalin for pain management. Now, she does not rely on this medicine anymore. At present, we only prescribe a daily low-dose folic acid supplement, vitamin B-complex, and low-dose aspirin [11] and will recheck the folate level soon in the follow-up clinic (Table 1). 

## 3. Discussion

Folate, or vitamin B9, is a water-soluble vitamin found in many foods whose metabolic derivatives are essential for the biosynthesis of purines and pyrimidines and protein metabolism [12]. Folate additionally plays a key role in the one-carbon metabolic pathway as it is necessary for the breakdown of homocysteine, an amino acid that can induce neurotoxicity when present at systematically elevated amounts [13]. Since hyperhomocysteinemia is an emerging comorbidity in ischemic stroke, folate intake and serum folate concentration are therefore hypothesized to be associated with cardiovascular disease risk through the modulation of homocysteine concentration [14]. Folate deficiency and/or *MTHFR* gene polymorphisms lead to hyperhomocysteinemia, which is a well-established risk factor for ischemic stroke in adult populations [14] as well as nontraumatic spinal cord infarction in children [15]. Thus, reductions in both folate and folate-associated nutrient (e.g., vitamins B6 and B12) levels within patient populations negatively impact the performance of the folate cycle, which subsequently impairs homocysteine metabolism [13]. 

Since methylenetetrahydrofolate reductase is the key enzyme for the biosynthesis of the active form of folate, we evaluated and selected three established pathogenic *MTHFR* gene variants, via ClinVar (https://www.ncbi.nlm.nih.gov/clinvar/ (accessed on 19 April 2022)), found in the Taiwan biobank database (https://taiwanview.twbiobank.org.tw/index (accessed on 19 April 2022)), which were verified by Sanger sequencing. However, no *MTHFR* gene variants were found to be applicable for this case (*MTHFR* gene (NM_005957.5: c.1969A>G, p.Ter657Arg (rs768434408); c.1304_1305del, p.Phe435fs (rs4846051); c.470C>T, p.Arg157Gln (rs121434295)). Our patient did not display elevated levels of homocysteine, nor did she possess three known pathogenic *MTHFR* mutations, and it was clear that the patient had a significant and isolated folate deficiency which can be treated via the supplementation of folic acid in order to prevent the onset of a new ASAS episode as well as to allow for the reparation of the patient’s injured spinal cord [4]. 

Next, we compared this case with similar reports involving the diagnosis of nontraumatic ASAS in adolescent populations (Table 2). We initially referred to similar cases fitting Zalewski’s criteria for spontaneous spinal cord infarction [16] and then excluded those with alternative acute myelopathy diagnoses, including those with intramedullary (e.g., inflammation, infection/post-infection, vascular anomalies, neoplasm, and toxin) and extramedullary causes (e.g., hematoma, abscess, compression, or spondylosis), and periprocedural acute myelopathy (e.g., aortic surgery). Finally, we recruited another 15 articles and an additional 27 cases with ages ranging from 10 to 19 years old (mean ± SD: 14.36 ± 1.82). A total of 28 cases are presented in Table 2, including 8 males (29%) and 20 females (71%). Some underlying medical conditions present within this population include obesity, asthma, early puberty, previously received spinal surgery, sickle-cell anemia, Down syndrome, and back pain. However, 19 cases did not have any underlying medical conditions of significance. Neurologically, most of the examined cases had sensory impairment (23/28, 82%), while all had various degrees of motor impairment. Additionally, 27 and 13 cases presented with bladder (27/28, 96%) and bowel dysfunction (13/28, 46%), respectively. Radiologic studies revealed lesions ranging from the lower medulla down to the conus medullaris, with the exception of two cases which did not receive the exam, and one which showed unspecified findings. Some potential etiologies include low serum folate levels, *MTHFR* gene variance, hemoglobinopathy, prothrombin gene variance, protein S deficiency, and fibrocartilaginous embolism (FCE), while 18 had unknown etiology. Later follow-up outcomes (mRS) revealed two cases of mortality; however, a single outcome was not available, with the remaining showing various degrees of neurologic improvement. 

Overall, female cases outnumbered males in our pooled data, and more than half of the cases examined did not have any detectable underlying diseases or identified etiologies. Furthermore, the clinical outcome varied greatly among these patients in the long-term follow-up. Nedeltchev et al. proposed that severe initial impairment (American Spinal Injury Association grades A and B) and being female were unfavorable predictors in adult cohorts, but this may not apply to pediatric cases of acute spinal cord ischemia syndrome [29]. Hence, nontraumatic ASAS is still a challenging issue and a critical topic in the clinical milieu. More case studies and extensive workups are still needed for further discovery of potential etiologies and risk factors in pediatric populations. 

## 4. Conclusions

The causes of nontraumatic ASAS remain mostly unknown, especially in pediatric populations. It is noteworthy that SCIWORA is one of the important differential diagnoses in ASAS. Since SCIWORA is mostly caused by hyperextension or flexion injuries of the pediatric spine, carefully evaluating the minor trauma and injuries is the first step in the approach of acute flaccid paralysis. After ruling out the traumatic etiology, extensive medical workups are necessary to explore the underlying pathomechanism of nontraumatic ASAS. Although more than half of the cases in our pooled data did not have underlying diseases or identified etiology, some still indicated potential predisposing factors, such as our case demonstrated here. Therefore, we suggest that folate deficiency is a potential risk factor in the adolescent ASAS population that needs to be medically identified and intervened upon early to improve the prognosis of these patients.

## Figures and Tables

**Figure 1 brainsci-12-01470-f001:**
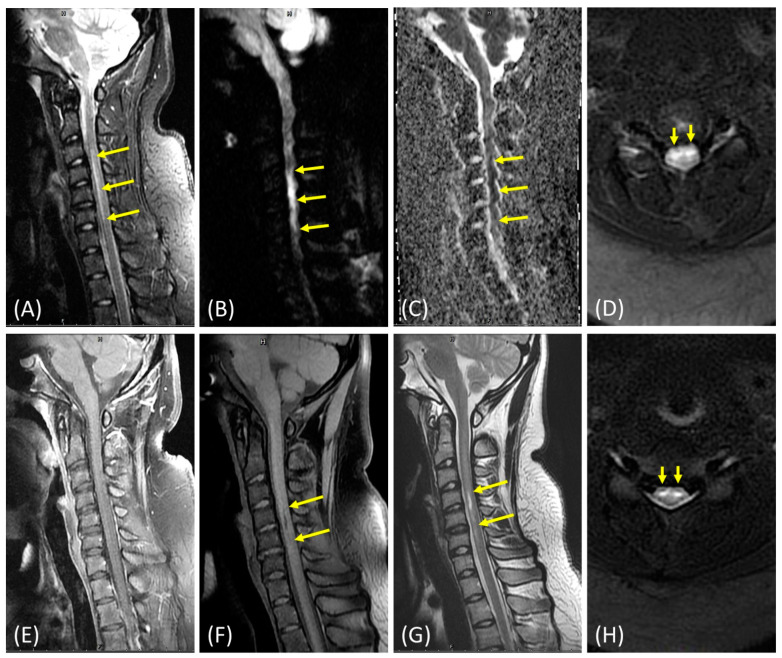
MRI reveals a hyperintensity in the patient’s cervical spinal cord between cervical vertebrae C3-6 on sagittal STIR images (**A**), which are associated with restricted diffusion abnormalities in the patient’s sagittal DWI (**B**) and apparent diffusion coefficient (ADC) images (**C**). Symmetrical elevated intramedullary signals in the anterior horns of the patient’s spinal cord (i.e. the “owl-eyes” sign for spinal cord infarction) were observed in the axial STIR images. Edema was also present at the same spinal cord level (**D**). No internal enhancements were detected in the sagittal T1-weighted contrast-enhanced images, which is inconsistent with the presence of infection and/or inflammation (**E**). A three-month follow-up MRI revealed a hypointensity between cervical vertebrae C4-5 in the patient’s sagittal T1-weighted fat-saturated (T1FS) images (**F**) and a hyperintensity in the sagittal T2-weighted images (**G**) due to myelomalacia. Axial STIR images (**H**) show a T2-hyperintensity in the anterior grey columns, in which the owl-eyes characteristic sign is observed as well. Arrows indicate the presence of abnormalities.

**Table 1 brainsci-12-01470-t001:** Case presentations of disease severity/score, clinical course, evaluations, prescriptions, and locations (Gantt chart).

	Category	Modified RS							S.E.	BH: 1.62 m		Evaluations		Prescriptions								Location			
Time Table		0	1	2	3	4	5	6	0-100	BW (kg)	BMI (kg/m^2^)	Laboratory Study	Image/Electrophysiologic Study	Pulse Therapy	A	B	F	L	P.I	P.O	P	I	W	H	S
**Week 1**									20	100	38.1	D1:Young stroke panel (B12, folate, HC) D1:Lumbar puncture	D1: Brain CTA/CTV D2: Brain/Spine MRI, NCV/EMG, echocardiography	D2~D6	TID	TID	QD	QD	QD						
**Week 2**									50	96.2	36.66	B12, folate, HC			TID	TID	1/2 QD	QD	QD						
*** Week 3**									62	93.5	35.63		D21: Carotid artery sonography		BID	TID	1/4 QD			QD					
**Week 4**									71	87.4	33.3				QD	TID				QD					
**Week 5**									75	89.9	34.26		D29: SSEP U/L limbs		QD	TID				QD					
*** Week 6**									84	89.4	34.06				QD	TID				QD					
*** Week 9**									NA	NA	NA				QD	TID				QD					
**Week 10**									88	86.9	33.11				QD	BID									
**Week 16**									NA	NA	NA		D96: Spine MRI follow-up		QD	BID					QD				
**Week 20**									90	93	35.44	B12, folate, HC	D120: NCV/EMG follow-up		QD	QD					QD				
**Week 24**									92	90.5	34.48				QD	QD	1/4 QD								

RS = Rankin Scale; S.E. = self-evaluation score; Pulse therapy = methylprednisolone 1000 mg/day; A = aspirin (100 mg), B = B1 (50 mg), B6 (50 mg), B12 (500 ug) complex; F = folic acid (5 mg), L = Lipitor (10 mg); P.I. = piracetam IV form (1000 mg); P.O. = piracetam oral form (1200 mg); P = pregabalin (75 mg); I = ICU; W = ward; H = home; S = school; HC = homocysteine; CTA/CTV = computed tomography arteriography/venography; MRI = magnetic resonance imaging; NCV/EMG = nerve conduction velocity/electromyography; SSEP = somatosensory evoked potential; U/L = upper/lower; NA = not available. * Week 3: D21, start rehabilitation program; * Week 6: transfer to other hospital for further rehabilitation; * Week 9: discharge from hospital and sent back to home and school. Gray degree of mRS: the darker implies the severer of the neurologic disability. Gray degree of location: the lighter implies that she can live a more normal life. TID: three times a day, BID: two times a day, QD: once daily. D1: day 1, D21: day 21, D29: day 29, D96: day 96, D120: day 120.

**Table 2 brainsci-12-01470-t002:** Comparison of cases with the diagnosis of nontraumatic ASAS in adolescence.

Study/ Year	No. case	Age/ Gender	Underlying diseases	Clinical Manifestations			Radiological lesions	Potential Etiology	Outcome at the latest FU (mRS)
				Sensory impairment	Motor impairment	Bladder/Bowel dysfunction			
**Our Case 2022**	**1**	**14Y/F**	**Obesity** **BMI:38.1kg/m^2^**	**Dysesthesia, temp., pain**	**Paresis: RU, LL**	+/+	C3─C6	Low serum folate level	1/6
**Sawada** [17] **2022**	1	10Y/F	Nil	Temp., pain	Paraparesis	+/−	C7─T2	Unknown	1/6
**Seo** [18] **2021**	1	12Y/F	Nil	Nil	Paraparesis	+/+	T12─CM	Unknown	1/6
**Bar** [15] **2017**	1	15Y/F	Asthma	Nil	Plegia: RL Paresis: LL	+/+	T11─L1	Unknown	3/6
	2	14Y/F	Nil	Temp., pain	Paraplegia	+/−	T7─T11	Unknown	1/6
	3	13Y/F	Nil	All modalities	Paraplegia	+/+	T8─T12	Unknown	3/6
	4	13Y/F	Nil	Nil	Paraplegia	+/+	CM	Unknown	3/6
	5	14Y/F	Early puberty	Temp., pain, LT	Plegia: LL Paresis: RL	+/−	CM	Unknown	2/6
**Spencer** [7] **2014**	1	12Y/F	Nil	All modalities	Paraplegia	+/+	T1─CM	Unknown	3/6
	2	14Y/M	Nil	Temp., pain	Paraplegia	+/−	T7─T12	Unknown	1/6
**Stettler** [19] **2013**	1	13Y/F	S/P spinal surgery	Dysesthesia, temp., pain	Plegia: RL Paresis: LL	+/+	T7─T9	*MTHFR,* Ho, c.677C>T	1/6
	2	13Y/F	Nil	Dysesthesia, temp., pain	Plegia: RL Paresis: LL	+/+	T3─T5	Unknown	1/6
**Márquez** [20] **2012**	1	19Y/M	Sickle cell disease	Nil	Quadriparesis	+/−	C2─C7	Hemoglobi-nopathy	2/6
**Sohal** [21] **2009**	1	16Y/F	Down syndrome	Not applicable^1^	Paraplegia	+/−	T5─T12	Unknown	2/6
**Nance** [4] **2007**	1	14Y/F	Nil	Dysesthesia, pain	Quadriplegia	+/+	Low medulla, C1─C7, T3	Unknown	5/6
	2	17Y/M	Back pain, palpitations	All modalities	Paraparesis	−/−	C2, T5─T9	*MTHFR,* CH, c.677C>T and 1298A>C	2/6
**Hakimi** [22] **2005**	1	15Y/M	Nil	Pain, LT	Paraplegia	+/+	Midthoracic level─CM	Prothrombin variant, He	3/6
	2	12Y/F	S/P spinal surgery	Temp., pain, PC	Paraplegia	+/+	Not performed^2^	Protein S deficiency	3/6
**Ramelli** [23] **2001**	1	15Y/M	Nil	Temp., pain	Paraplegia	+/+	T5─T6	Protein S deficiency	3/6
**Wilmshur st** [24] **1999**	1	14Y/F	Nil	All modalities	Paraplegia	+/−	T9─CM	Unknown	4/6
	2	15Y/F	Nil	Temp., pain, PC	Paraplegia	+/−	Anterior+Right posterior cord	Unknown	3/6
	3	14Y/M	Nil	Dysesthesia, temp., pain	Paraplegia	+/−	Anterior distal thoracic cord	Unknown	3/6
	4	16Y/M	Nil	Pain	Paraparesis	+/−	T3─T6	Unknown	3/6
	5	15Y/F	Nil	Temp., pain	Paraplegia	+/+	T9─CM	Unknown	4/6
**Yousef** [25] **1998**	1	14Y/F	Obesity BMI:33.6kg/m^2^	Pain, LT	Paraplegia	+/−	Unspecified	FCE	6/6^3^
**Tosi** [26] **1996**	1	16Y/F	Nil	Temp., pain, LT, PC	Paraplegia	+/−	T11-L1	FCE	No description
**Toro** [27] **1994**	1	16Y/F	Nil	All modalities	Paraplegia	+/−	Not performed	FCE	6/6^4^
**Vandertop** [28] **1991**	1	17Y/M	Nil	Dysesthesia, temp., pain	Quadriparesis	+/−	C1-T3	Unknown	2/6

**Abbreviations:** Y: years old, F: female, M: male, Temp.: temperature, LT: light touch, PC: proprioception, C: cervical, T: thoracic, L: lumbar, CM: conus medullaris, S/P: status post, Ho: homozygous, He: heterozygous, CH: compound heterozygous, FCE: fibrocartilaginous embolism. **Notes:** ^1^ Not applicable: accurate sensory examination of the lower limbs was difficult due to learning difficulties and distress present in this patient, however, sensation to light touch, vibration, and joint position in the lower limbs appeared to be intact. ^2^ Not performed: due to extensive metallic hardware after corrective posterior spinal fusion. ^3^ The patient left the hospital against medical advice on the third day of hospitalization. She died 7 days later from a massive pulmonary thromboembolus proven by autopsy. ^4^ The patient died unexpectedly as a result of bronchial aspiration 6 weeks after disease onset.

## Data Availability

All the data are contained within the article.

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
