# Peer review of "The Role of Folate Deficiency as a Potential Risk Factor for Nontraumatic Anterior Spinal Artery Syndrome in an Adolescent Girl"

_brainsci, 2022, doi:10.3390/brainsci12111470_

Round 1
Reviewer 1 Report
Thank you for the privilege of reviewing the manuscript on nontraumatic anterior spinal artery syndrome (ASAS) in an adolescent girl. I think the article is interesting for readers of brain sciences.
I recommend to expand the introduction.
Please answer some questions:
How can you differentiate ASAS from SCIWORA?
How did you discern ASAS from Guillain Barre syndrome?
Why did you administer aspirine instead of heparine in this patient?
Did you perform MR-angiography in your patient?
Please can you describe outcome in more details in your patient?
Did you receive informed consent from the patient and her family?
Reviewer 2 Report
This manuscript is centered on a case report, describing the sudden onset of anterior spinal cord injury syndrome in an adolescent girl.
I agree that this syndrome is extremely rare, particularly in children, especially when no profound risk factors are evident (i.e. aortic surgery, trauma) and the relevant pathogenetic mechanisms are not established.
However, the diagnostic work up is inadequate, especially when we attempt to correlate this entity with folate deficiency. In this age group, although not common, we should exclude the possibility of SCIWORA, and a more detailed medical history should be recorded. Also, a more extensive work up should include dynamix X-rays and CT of the cervical spine to rule out a spinal injury. Moreover, you mention that 'a young stroke panel was also used to extensively survey for metabolic (e.g Fabry disease) and autoimmune diseases (e.g. the presence of anti-AQP4 IgG and anti-MOG IgG), along with hypercoagulable disorders.' As the anterior spinal artery syndrome is associated with thrombotic and embolic disesaes, a more comprehensive presentation of your relevant results is needed. Finally, you have not perfomed a DSA of the spinal arteries, or at least an MRA, to image the offending pathology to the anterior spinal artery. You have mentioned that ' The patient received a five-day course of high-dose intra- venous methylprednisolone to ameliorate the spinal cord edema, aspirin for the anticoagulation, and folic acid supplementation for the marked folate deficiency'. As far as thrombosis or embolus of the ASA was not evident, I am reluctant about the use of aspirin. Apart from that, you have not performed a follow-up MRA or DSA of the cervical spine, to have a feedback for the patient. For example, what should be the duration of apirin therapy? Are there any convincining evidence that the neurological imrovement was due to folic acid supplementation or to high-dose intra- 80 venous methylprednisolone?
Reviewer 3 Report
The article by Hu et al. "Folate Deficiency as a Potential Risk Factor for Nontraumatic Anterior Spinal Artery Syndrome in an Adolescent Girl" covers a potentially interesting and emerging topic related to the rare ASAS syndrome. In this sense, this remains to be potentially interesting for the Brain Sciences readers. I regard the main point of this paper as highly attractive as well as the results are clearly presented. The text does not contain any major errors, therefore I have some minor comments and recommendations:
1. There is a need to provide slightly more expanded introduction regarding the pathogenesis of disease with epidemiology
2. The figure in introduction section should be added
3. Following references should be added and properly cited within the main text:
- Kubaszewski Ł, Wojdasiewicz P, Rożek M, Słowińska IE, Romanowska-Próchnicka K, Słowiński R, Poniatowski ŁA, Gasik R. Syndromes with chronic non-bacterial osteomyelitis in the spine. Reumatologia. 2015;53(6):328-36. doi: 10.5114/reum.2015.57639.
- Xing W, Zhang W, Ma G, Ma G, He J. Long-segment spinal cord infarction complicated with multiple cerebral infarctions: a case report. BMC Neurol. 2022 Sep 22;22(1):362. doi: 10.1186/s12883-022-02888-8.
- Tykocki T, Poniatowski ŁA, Czyz M, Wynne-Jones G. Oblique corpectomy in the cervical spine. Spinal Cord. 2018 May;56(5):426-435. doi: 10.1038/s41393-017-0008-4.
- Hsu JL, Cheng MY, Liao MF, Hsu HC, Weng YC, Chang KH, Chang HS, Kuo HC, Huang CC, Lyu RK, Lin KJ, Ro LS. A comparison between spinal cord infarction and neuromyelitis optica spectrum disorders: Clinical and MRI studies. Sci Rep. 2019 May 15;9(1):7435. doi: 10.1038/s41598-019-43606-8. PMID: 31092838; PMCID: PMC6520381.
4. In some places the use of English could be improved on.
Completing this gaps will have an impact on the understanding the aim of the study and, from my point of view, is absolutely necessary.
Round 2
Reviewer 2 Report
Dear Authors,
I still raise concerns about the scientific soundness of your article. However, this paper could be published if the following requirements are fullfilled.
You have mentioned to your reply that 'neither ideal biomarkers nor convincing evidences that prove the independentefficacy of folic acid supplement as well as high dose methylprednisolone on the neurologic outcome. The patient may benefit from the summation of all our treatments, including other supportive care (e.g., rehabilitation program), but we can’t discriminate each contribution from all these treatments/managements.I believe that each of them hasimpacts on the outcome to some degree.' Because of that, you have to modify the title of your manuscript, delineating that the role of folate deficiency is questionable. Moreover, as I have mentioned, you have not performed a DSA of the spinal arteries, or at least an MRA, to image the offending pathology to the anterior spinal artery. This should be mentioned as a potential factor that limits the valuability of your study.
